# EREC: Enhanced Language Representations with Event Chains

Huajie Wang [1,2] and Yinglin Wang [1,*]

1 School of Information Management and Engineering, Shanghai University of Finance and Economics, Shanghai 200433, China
2 School of Management Science and Engineering, Shandong University of Finance and Economics, Jinan 250014, China
* Correspondence: wang.yinglin@shufe.edu.cn

**Abstract:** The natural language model BERT uses a large-scale unsupervised corpus to accumulate rich linguistic knowledge during its pretraining stage, and then, the information is fine-tuned for specific downstream tasks, which greatly improves the understanding capability of various natural language tasks. For some specific tasks, the capability of the model can be enhanced by introducing external knowledge. In fact, these methods, such as ERNIE, have been proposed for integrating knowledge graphs into BERT models, which significantly enhanced its capabilities in related tasks such as entity recognition. However, for two types of tasks, commonsense causal reasoning and predicting the ending of stories, few previous studies have combined model modification and process optimization to integrate external knowledge. Therefore, referring to ERNIE, in this paper, we propose **e**nhanced language **r**epresentation with **e**vent **c**hains (EREC), which focuses on keywords in the text corpus and their implied relations. Event chains are integrated into EREC as external knowledge. Furthermore, various graph networks are used to generate embeddings and to associate keywords in the corpus. Finally, via multi-task training, external knowledge is integrated into the model generated in the pretraining stage so as to enhance the effect of the model in downstream tasks. The experimental process of the EREC model is carried out with a three-stage design, and the experimental results show that EREC has a deeper understanding of the causal relationship and event relationship contained in the text by integrating the event chains, and it achieved significant improvements on two specific tasks.

**Keywords:** external knowledge; event chains; commonsense causal reasoning; story ending prediction





## 1. Introduction

BERT (Bidirectional Encoder Representations from Transformers) [1] is designed to pretrain deep bidirectional representations from unlabeled text by jointly conditioning left and right contexts. BERT is generally a two-stage model, and it first provides a general pretraining model with rich linguistic knowledge and then solves downstream tasks via a fine-tuning mode. It achieved state-of-the-art results on as many as 11 different NLP tasks and has been widely used ever since.

After continuous developments, many studies attempted to break the traditional two-stage process and use some other methods to enhance the ability of BERT. Among them, there are two mainstream approaches. One is a three-stage process approach in which an additional stage is added between the pretraining stage and fine-tuning stage, and an additional corpus is used for incremental training in the extra stage. The other is the transform model approach in which the model's structure is transformed and additional training tasks are designed in the traditional pretraining process of BERT.

The well-known three-stage process approach was proposed by Li et al. [2], who added a second stage of pretraining in the middle of the traditional BERT pretraining stage and the fine-tuning stage. The new stage is used to train causal task-related supervised tasks with respect to external causal knowledge. By utilizing three specific transfer tasks,

including natural language inference, emotion classification, and action prediction, the prediction accuracy of the Story Cloze Test (SCT) is improved.

Transform models, such as the models proposed in references [3–5], attempt to incorporate external knowledge into pretrained language models. These models focus on the integration of a knowledge graph and a language model. One study of this approach was conducted by Zhang et al. [4] who integrated knowledge into BERT. In their research study, a unified vector space is formed by using a large-scale text corpus and knowledge graph to train an enhanced language representation model ERNIE. This model can simultaneously use lexical, syntactic, and knowledge information.

The above studies show that compared with the traditional two-stage design, the three-stage process and the transform model are able to enhance the effect of BERT in some specific tasks. Inspired by the above approaches, we propose a three-stage model, **e**nhanced language **r**epresentation with **e**vent **c**hains (EREC), which is a combination of the above two approaches. We apply the model in commonsense causal reasoning tasks and story ending prediction task. The EREC model contains the following: (1) a pretrained BERT model [1] in stage 1; (2) a denoising autoencoder (DAE) [6] in stage 2—similarly to the model structure proposed by [4], in which the event chains contained in a text corpus are extracted and aligned with the text. Then, it is unified with the graph network's representation; this enables the model to have incremental pretraining capabilities. With this synchronous alignment of text and event chains, EREC achieves the effective fusion of linguistic representation and event association. Finally, there is a (3) fine-tuned component in stage 3: the model produced in stage 2 is fine-tuned for specific downstream tasks.

The second stage is the core of EREC, which includes modules for preprocessing the corpus, extracting event chains, aligning text, fusing graph network representations, and incremental training. Since the core of the model proposed in this paper includes event chains, which are extracted from the corpus containing causal or event relations, the model is then evaluated in two specific downstream tasks: choice of plausible alternatives (COPA) [7] and story cloze test (SCT) [8]. The experimental results show that EREC significantly outperforms the state-of-the-art BERT model on the task by taking full advantage of the events' relations.

In summary, this paper contributes to the following areas:

- This research proposes an enhanced language representation model, named EREC, which incorporates the semantics of the text and the causal event relationships contained in it, and such relationships are extracted and used as event chains to enhance the ability of the model to understand the text's information.
- A three-stage process model is used, and an incremental training process is conducted in the second stage to integrate the above extracted event chains and to achieve better results in the two downstream tasks.
- In the process of merging the event chains into the embeddings, multiple experiments are compared. These experiments are conducted in the form of *network + algorithm* combinations in which three graph networks and two different algorithms (Deepwalk and Line) are compared. By using comparisions, a particular setting was found to be more effective for two downstream tasks.

## 2. Related Work

Many efforts are devoted to improving the semantic understanding and representation capabilities of pretrained language models so as to accurately capture linguistic information from the text and to utilize it for specific NLP tasks. From the perspective deciding whether to change the model's structure, this type of enhancement method can be divided into two classes, i.e., knowledge fusion and incremental training.

A typical work of knowledge fusion is the ERNIE model proposed by Zhang et al. [4], which integrates the knowledge graph into BERT. In the *fine-tune* stage, the knowledge graph is extracted by the *TagMe* tool before entity linking, and ERNIE only verifies *entity typing* and *relation classification* and does not extend to other NLP tasks. In addition,

there has also been a great deal of research in knowledge augmentation, such as [9–13], both of which have made improvements to models for downstream tasks and achieved effective enhancements.

For incremental training, Gururangan et al. [14] proposed useing domain-related corpora, task-related corpora, or a combination of them to continue incremental pretraining on the basis of BERT, and the obtained model achieved significant improvements in a variety of NLP tasks. Furthermore, Gu et al. [15] combined the domain-related corpus with the task-related corpus and adopted incremental training to enhance the capability of the model. It is worth noting that Gu et al. designed an algorithm to mark task-related keywords in the corpus, such as *like* and *hate*, which play important roles in sentiment classification tasks, while Clark et al. [16] extracted the keywords in the text and presented them in the form of event chains.

In this paper, we combine the above two methods to integrate external knowledge into the language model during the process of incremental training. Nouns and verbs are usually used to understand semantic relationships [17,18], so they are also chosen as external knowledge; for example, event/casual relations implied in the nouns and verbs of a text corpora and those explicitly expressed in a network can be used to enhance the reasoning ability of the model. Among the causal-relation-mining studies, Luo et al. [19] showed that a complete pair of causal statements usually contains a pair of words with causal relations. For example, in the two statements, *I knocked on my neighbor's door and my neighbor invited me in*, *knocked* and *invited* have a causal relationship. Luo et al. also extracted a large number of causality pairs similar to *knock* and *invite* from a large data set and constructed a rich causality network called causalNet. Li et al. [20] conducted a further study on the basis of causalNet and provided a corpus with causal characteristics (CausalBank) and a large lexical causal knowledge graph (cause effect graph). Li et al. designed intermediate tasks to incrementally train BERT using the CausalBank corpus and achieved good results in the COPA task.

In addition to the above studies, the graph's embedding may have an impact on the results of tasks. Among the generation algorithms for graph embedding, *deepwalk* [21] is more common, which is derived from word2vec [22] and its core comprises random walk. However, *deepwalk* only works on undirected graphs and not directed graphs, and it does not take into account the weight between nodes. Tang et al. [23] propose a large-scale network coding model, *Line*, in which the calculation of second-order similarities is used on directed or undirected graphs, and the weight between nodes is taken into account. In this paper, we use these two algorithms to conduct comparative experiments.

## 3. Approach and Model Structure

In this section, we present EREC's model structure in Section 3.1 and the details of integrating the event chains in the corpus and network embedding into the model, which includes events' extraction in Section 3.2, network embedding in Section 3.3, and tokens and events Alignment in Section 3.4.

### 3.1. Model Structure

As shown in Figure 1, we adopt a three-stage experimental model, EREC, in this paper, including a pretrained BERT model [1] in stage 1, incremental training in stage 2, and specific fine-tuning tasks, COPA and SCT, in stage 3. The core of EREC is the external knowledge fusion and incremental training in the second stage. In this stage, the underlying semantics of the input tokens are derived primarily from the textual encoder (T-Encoder) of the model, which is an encoder similar to BERT and sums the token embedding, segment embedding, and positional embedding for each token to compute its input embedding. Another encoder, a knowledgeable encoder named K-Encoder, is contained in stage 2, which is similar to ERNIE and takes the event and event embedding as inputs, where the details of event embedding are described in Section 3.3. Then, multi-tasks, including next sentence prediction (NSP), masked language model (MLM), and denoising autoencoder

(DAE), are trained to integrate their knowledge with the semantics derived from T-Encoder, where the details of multi-task training will be described in Section 4.2.

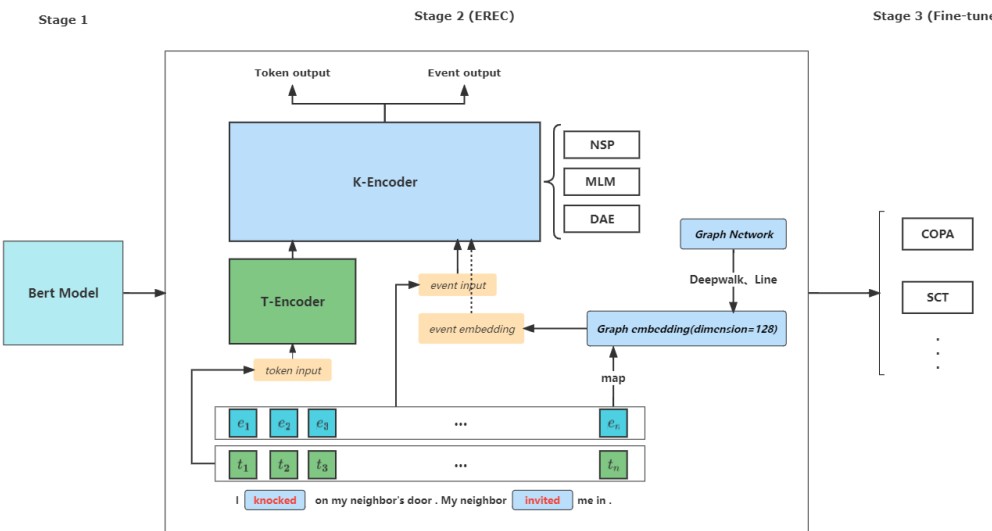

**Figure 1.** Model architecture and three-stage process.

The generalization ability of the model is improved by introducing noise. This is based on the fact that partially corrupted data have the same expressive power. Therefore, a good representation can capture the stable characteristic structure of its observed input distribution. For high-dimensional redundant inputs (such as text), similar stable structures can be collected through the combination of multiple dimensions, so the sentence can be repaired from partial corruption. For example, "a thief entered the store and stole the necklace", after masking the word "steal", we may still "guess" the word or some word that has similar meaning. For example, in this paper, DAE encodes masked event sequence, then need the decoder to reconstruct the origin event sequence from the sequence embedding. This property is similar to the ability of humans to recognize partially occluded or damaged sentences, which means that humans can abstract higher-level concepts from various association of things, and even if some forms disappear, they can still make identifications from them.

Thus, we add a noise-processing mechanism at K-Encoder, similarly to the denoising autoencoder (DAE) [6]. As shown in Figure 2, the specific step is to obtain $\tilde{x} \sim q_D(\tilde{x}|x)$ by random masking with respect to input $x$. $q_D$ refers to the mean of a stochastic mapping, where $D = \{x_1, x_2, x_3 \ldots x_n\}$ is the input sequence. In the experiment, the following masking form is used. For each input x, given a random mask proportion $v$, the masked values are completely removed, and the rest remained unchanged. Masked input $\tilde{x}$ is mapped to $y$ by the autoencoder, $y = f_\theta(\tilde{x}) = s(W\tilde{x} + b)$, and further reconstructs $z = g_{\theta'}(y) = s(W'y + b')$. We use the *traditional squared error* (TSE) loss during training by building a reconstruction error function $L$

$$L(x, z) = ||x - z||^2 \tag{1}$$

That is, the reconstructed $z$ should be as close as possible to the original sample input $x$.

Similar to *denoising autoencoder* (DAE), after the random mask, the token's sequence $\{t_1, t_2, t_3, t_4 \ldots t_n\}$, where $t_i (1 \leq i \leq n)$ represents the i-th token in the sequence, is mapped to event sequence $\{e_1, e_2, e_3, e_4 \ldots e_m\}$, where $e_j (1 \leq j \leq m)$ represents the j-th event in the sequence. In order to obtain the masked input, the following strategies are used for the original input tokens:

(1)　The probability of 80% remains unchanged;

(2)　A probability of 10% is used to mask the *event* aligned with the *token*, and it is set to −1, which aims to make the model to predict the masked events;

(3)　A probability of 10% is used to replace the *event* aligned with the *token* with other events. For example, the event's id is replaced, corresponding to *knock* in the original corpus with the event id corresponding to *strike*, which is intended for training the model in order to correct errors.

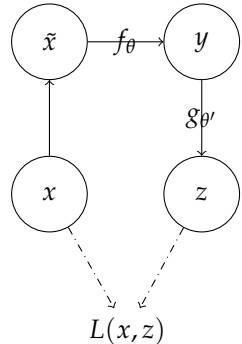

**Figure 2.** The process of denoising autoencoder (DAE).

### 3.2. Event Extraction

To realize the enhanced language representation with event chains, it is necessary to perform a series of tasks, including sentence segmentation, word segmentation, part-of-speech tagging, and event extraction. The quality of work in each stage will affect subsequent tasks. Since the focus of this paper is to solve the fusion of the language model and event chains, we directly use the method proposed by Chambers [24] to extract narrative event chains. The steps of the process are as follows.

Firstly, we divide the COPA statements mentioned in Luo et al. [19] into two lines and take them as the original corpus for subsequent processing; an example is provided as follows.

> *I knocked on my neighbor's door.*
> *My neighbor invited me in.*

Then, we obtain the event chains extracted from the corpus by using the method of Chambers [24] so that the keywords, keyword prototypes, and their positions in each sentence are marked. The **events** extracted from the above example are as follows.

> *Events:*
> *knocked / knock / verb_pos=(0,1) / type=normal / subj=I / obj=door / iobj=None*
> *invited / invite / verb_pos=(1,2) / type=normal / subj=neighbor / obj=me / iobj=None*

Taking "knocked" as an example, the extracted items are separated by "/". The meaning of each item in the order from left to right are as follows: (1) the verb of the event in the original text, which is "knocked" here; (2) the prototype of the verb: knock; (3) *verb_pos* indicates the position of the word in the original text, and its value (0, 1) indicates the first position of the word in line 0; (4) *type = normal* indicates that the extraction result is normal and the syntax is normal; (5) *subj = I* indicates that the subject of the event is "I"; (6) *obj = door* indicates that the object of the event is "door"; *iobj* indicates the indirect object of the verb. Here, *iobj = None* indicates that there is no indirect object. In this paper, since we focus on the key relationships between events, we only use the first three items. Thus, the verb in the sentence is treated as the event in this paper.

### 3.3. Network Embedding

In order to represent relationships with respect to causality, we use the Causality Network (CausalNet) provided by Luo et al. [19], which is extracted from the corpus with

causal characteristics. In CausalNet, the relations between events are mined and built into a graph network, where nodes correspond to events and edges correspond to relations. The network obviously contains rich causal relationships.

Each row in CausalNet is in the form of (*word, word, number*), which corresponds to an edge (*node, node, weight*) in the network, where *node* represents *event* and *weight* represents the degree of association between two events. For example, an edge (*white, congratulate, 3*) indicates that the strength or degree of causal association between "white" and "congratulate" is 3. From CausalNet, we extract different node information to construct an ***events vocabulary***.

In order to construct gragh embeddings, the following two algorithms are used: (1) **LINE with second-order proximity [23]**—the input of this algorithm is a directed graph network and its edges are in the form of (*word, word, number*). The algorithm is applicable to both directed and undirected graphs, and the weights between nodes in the network can be incorporated into the graph's embedding. (2) **DeepWalk [21]**: The edges of the network have no weight and their form is (*node_index: x, node_index: y*). Therefore, we remove the weights in the CausalNet network and the words of each node are converted into their indexes in the events vocabulary as the *node_index* here; thus constructing an unweighted network that conforms to the deepwalk input form.

By using these two algorithms, a 128-dimensional embedding vector is generated for each node in the causality network.

### 3.4. Tokens and Events Alignment

A key part of the integration between the language model and event chains is to align tokens and events, which is the basis of incremental training in this paper. Therefore, it is necessary to mark the events in the original text corpus. To perform this, the location of events (marked in the text corpus) is used, and the events (verbs) are wrapped with the 'sepsepsep' separator. After transformations, for example, the previous sentence, 'I knocked on my neighbor's door', becomes the following form.

<div align="center">

*I **sepsepsep** knocked **sepsepsep** on my neighbor's door.*

</div>

The event extraction process (mentioned in Section 3.2) uses OpenNLP to perform preliminary sentence and word segmentation operations on the text. In order to meet the input format of the BERT model, it is necessary to perform more refined word segmentation on the text.

For example, the model used in this paper is *bert-base-uncased*, its vocabulary size is 30,522, and the letters are all in lowercase form. We use BERT FullTokenizer to divide all the corpus into tokens contained in this vocabulary.Then, there will be one problem: after using BERT for word segmentation, each sentence is divided into a sequence of tokens, that is, $TS = \{t_1,t_2,t_3,t_4\ldots t_m\}$. The events corresponding to tokens in the sequence $TS$ form another sequence: $ES = \{e_1,e_2,e_3,e_4\ldots e_n\}$. In order to align the events to the corresponding tokens. We define an alignment function as $f_{alignment}(\{t_1,t_2,t_3,t_4\ldots t_m\}) = \{e_1,e_2,e_3,e_4\ldots e_n\}$ to perform this step. Thus, tokens and events are semantically aligned. Please note that the number of events is different from the number of tokens. This is because we only use verbs as indicators of events in this paper; moreover, a verb will form multiple morphemes after word segmentation via the BERT FullTokenizer. For example, *resuscitate* (regarded as an event) is aligned with *res, ##us, ##cit, ##ate*.

The first idea will be adopted in subsequent experiments in this paper. After word segmentation, the text can be converted into the id sequence in the BERT vocabulary. For example, the id of *knocked* in the BERT vocabulary is 6573. It is worth noting that a large number of *sepsepsep* separators have been inserted into the text to mark events. As the maximum id of the vocabulary of BERT is 30,521; thus, we set the id of 'sepsepsep' as 30,522. Actually, it can be skipped and remain unused in the model. Before subsequent processing, we record the correspondence from the event between the two delimiters to the event in the *events dictionary* generated in Section 3.3, such as *knocked → knock → 2404*, and

we align other non-event tokens to #UNK#. After the above process, as shown in Figure 3, two sequences (token sequence and event sequence) are generated from each sentence in the text, which is ready for subsequent processing. *MASK* in Figure 3 means to mask or replace the original event with other events. For details, please refer to Section 4.

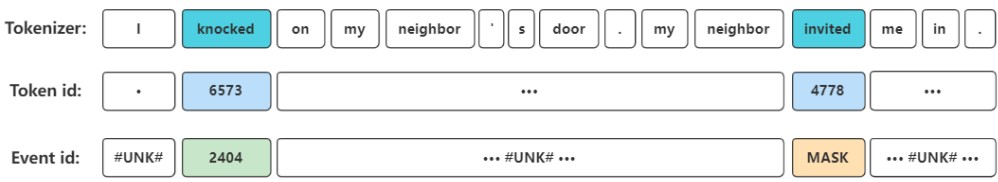

**Figure 3.** Alignment of an input example.

## 4. Experiments

In this section, the details of the entire three-stage process are presented, particularly the pretraining of the second-stage EREC model, which is also a BERT incremental pretraining step. Additionally, the first stage follows the procedure of the BERT model [1], which is not repeated here, but the publicly released pretrained BERT model, *bert-base-uncased*, is used.

### 4.1. Pretraining Dataset

We use the CausalBank provided by Li et al. [20] as our pretraining dataset. Causal-Bank is a causal corpus containing 314 million CE (cause–effect) pairs.

CausalBank contains different causal connectives, which are called *patterns* in [20], and they mainly include *as, as a consequence/result of, as long as, because, because of, caused by, due/owing to, in response to, on account of, result from*, etc. Each pattern constructs a file, and each line in the file represents a complete causal statement in the form of PEC (pattern—effect—cause). For example, *because—I am very sad—I lost my phone*.

Considering that the size of the corpus is too large, as mentioned by Ding et al. [25], a smaller number of samples with strong event correlations tend to train the model more effectively than a larger number of irrelevant samples, which shows that the sample quality is more important than the sample size. In order to obtain training examples that are similar to COPA's form from these selected sentences of CausalNet, we transform the sentences in the following manner. We automatically remove the causal connectives from the compound sentence, and we extract the cause clause and the effect clause. Then, each pair of the two clauses is marked as a "from cause to effect" or "from effect to cause" pattern according to the order of the two clauses. For example, after removing the causal connectives "because …so …", the sentence "because I lost my phone, so I am very sad." becomes a pair comprising two sentences, (*I lost my phone. I am very sad.*), which will be marked as the "from cause to effect" pattern, similarly to COPA. On this basis, the corpus will be further processed according to Section 3 and will be ready for subsequent processing.

### 4.2. Multi-Task Training

Similarly to ERNIE, the EREC model performs a multi-task procedure in which the tasks of next sentence prediction (NSP), masked language model (MLM), and denoising autoencoder(DAE) are trained. For the NSP task, the *cause* in the corpus is taken as *tokens_a*, the *effect* corresponding to *tokens_a* is taken as *tokens_b* with a probability of 50%, and a sentence is randomly selected from the corpus as *tokens_b* with a probability of 50%.

For the core task DAE, we align the sentences in the CausalBank with the extracted event sequence obtained from the text, and we use the mask strategy described in Section 3.3 to process the original input event sequence. The training target is composed of two parts: predicting the masked event and correcting the replaced event. There are two key stages in this process.

(1) **Event Number**

Each event sequence should usually contains at least two event keywords, and a causal relationship is expected. However, as an event sequence containing less than two event keywords can enhance semantic understanding, and it is still added to the incremental training process.

(2) **Batch-size**

After data preprocessing, the number of events in each text has been fixed, so the batch size has become a key factor affecting candidates of events, which is the set of all events in a batch of data. In the DAE task, the model will take the candidates as an option list to predict and correct the masked and replaced events, respectively. If the batch-size is too small, there will be few events in the candidates, resulting in insufficient causal relationship with respect to model fusion; if the batch-size is too large, incremental training requires fewer training steps, which affects the search for optimal model parameters. Thus, a suitable batch-size is crucial.

*4.3. Parameter Settings and Training Details*

Given the characteristics of the CausalBank corpus, we set the maximum sequence length to 40 and found a suitable batch size for the DAE task, which is set to 32 in our experiments. We use BertAdam [1] as the optimizer, and the initial learning rate is set to $5 \times 10^{-5}$. The value "$5 \times 10^{-5}$" refers to the value of the learning rate as the training parameter of the model. It is provided in scientific notation and equals 0.00005. For the rest of the super parameters, we use the same parameters as those used in BERT.

The sum of the losses of the three tasks, NSP, MLM, and DAE, is used as the loss value for model training. Whenever training reaches a certain number of steps, for example, 10 steps, a prediction is made on the test set of a specific downstream task. Experiments show that, within an epoch, the overall accuracy rate increases first and then decreases, and we save the model's parameters when the accuracy rate reaches the highest point; thus, we can meet our experimental needs without a large amount of data. Based on this, subsequent experiments only use 30 K sentences for incremental training, and the experiments use an early stopping strategy when the prediction accuracy reaches the peak during training. Then, the model's parameters at the highest accuracy are saved for subsequent work.

*4.4. Fine-Tune for Specific Tasks*

The COPA task is used during the fine-tune stage. We divide all the corpuses contained in COPA (*COPA-All*), which contains 1000 commonsense causal reasoning questions, into a training set called *COPA-Train* and a testing set called *COPA-Test* in a ratio of 7:3, respectively. *COPA-Test* is also used as the test set in the second stage. Then, the model that achieves the highest prediction accuracy in the second stage is reserved for further fine-tuning in the third-stage. In the fine-tuning stage, the maximum sequence length is also set to 40.

**5. Results and Analysis**

Our proposed EREC model is evaluated by its performance on downstream tasks. The corpus CausalBank is used for COPA tasks in this section, which requires the model to choose the correct cause or effect from two candidates. In order to expand the application fields of the EREC model, in addition to the causal relationship, the event relationship is also incorporated into the model to verify the ability of the model in the Story Cloze Test (SCT).

*5.1. Model Performance*

Three different models are used and compared for the same downstream task, COPA:

(1) **BERT-base [26]**;

(2) **CausalBert-base:** The model proposed by [20] designs an intermediate task for training on the basis of BERT, and its test data include 100 randomly sampled input examples from the model's training data and 100 randomly sampled examples from the development set of the COPA dataset.

(3)   **EREC:** The model proposed in this paper.

For COPA tasks, these models are trained with a margin-based loss [26]. The model EREC proposed in this paper transforms the model's structure while performing incremental training, and it integrates the external causal network; we obtain an improvement of 5.2%, from 75.4% to 80.6%. The causal network of the EREC fusion shown in Table 1 is causalNet, and the deepwalk algorithm is used to generate the embedding.

**Table 1.** Results on *COPA-Test*.

| Method | Accuracy (%) |
|---|---|
| BERT-base [26] | 75.4 |
| CausalBert-base [20] | 78.6 |
| EREC(Ours) + causalNet | **80.6** |

### 5.2. Ablation Experiment

To better verify key factors in the EREC model, we perform an ablation experiment. In the process of incremental training, after removing event chains from the training data and removing the training task but retaining the rest as the same as that of EREC, we obtain a model without event data, and we call it EREC (No events).

The results of the ablation experiment in Table 2 show that the model that is integrated with event chains in incremental training can better improve the results of downstream tasks and further verify the core role of event chains in EREC.

**Table 2.** Ablation Experiment.

| Method | Accuracy (%) |
|---|---|
| EREC (No events) | 76.3 |
| EREC + causalNet | **80.6** |

### 5.3. Comparative Experiment

As proved by the experiments in the previous section, the key to EREC lies in the integration of event chains and the causal graph network during the second-stage incremental training process. Therefore, the comparative experiments in this section mainly focus on the generation of the graph network and network embedding. Three different networks are used:

(1)   CausalNet [19]: for the convenience of explanations, it is abbreviated as *CNT* in this paper;

(2)   *CEG* [20] is a causal network, such as CausalNet, and both are weighted by co-occurrence frequency;

(3)   Li et al. [27] provided us with an event network, which processes the extracted events into the form of predicate-GR [16]. For example, the mapping of *knock* in the event vocabulary formed by the network is *knock → knock_subj → 55,450*, so it is abbreviated as *PGR* in the experimental conclusion.

Table 3 shows the experimental results of different combinations of graph networks and network embedding generation algorithms on the *COPA-Test* set.

Three different graph networks are used for fusion in the EREC incremental training process. Compared with BERT-base, the results of the obtained models in the COPA task significantly improved. The model using CNT achieves the best performance. Using CEG does not obtain the best result. We notice that there are more nodes and edges in CEG, and the average weight of the edges far exceeds CNT, which has a value of 10.54. Therefore, we believe that in the case of the limited corpus, using a smaller-scale network CNT can better integrate causality into the model, and CEG is suitable for a larger-scale corpus. PGR covers a wider range of event relations, and the experimental results show that it also

has certain gains for causal reasoning tasks. In addition, for the same network such as CNT, compared with deepwalk, the Line algorithm additionally considers the weight and directionality between events and obtains a better result. This may be because, during the training process, the embedding generated by Line contains more accurate causality and, thus, has a better gain on the effect of the model.

**Table 3.** Results on the *COPA-Test*.

| Algorithm<br>Graph Network | Line | Deepwalk |
|---|---|---|
| CNT | **81%** | 80.6% |
| CEG | 77.6% | 79.3% |
| PGR | 78% | 77.6% |

In order to further improve the result of the model, we replaced BertAdam with Adam [28] or AdamW [29], and the COPA task is repeated in the third stage of the fine-tuning process. The experimental results with different optimizers on the *COPA-Test* dataset are listed in Table 4. Zhang et al. [30] mentioned that the gradient bias correction is omitted in the BertAdam optimizer, which is not beneficial for the fine-tuning of the model, especially on small datasets. In the early stage of training, the model will continue to oscillate, which will reduce the efficiency of the entire training process, slow down the speed of convergence, and cause fine-tuning instability.

**Table 4.** Experimental results with different optimizers when fine-tuning the *COPA-Test* dataset.

| Network<br>Optimizer | CNT/Line | CEG/Deep | PGR/Line |
|---|---|---|---|
| BertAdam [1] | 81% | 79.3% | 78% |
| Adam [28] | 81% | 80% | 81% |
| AdamW [29] | **82.6%** | 80.3% | 81.6% |

As shown in Table 4, the results in Table 3 are selected, with better results from different experimental settings. For example, for network CEG, we select deepwalk as the generation algorithm for further experimental analyses.

The results of the comparative experiments once again proved the point of view proposed by Zhang et al. [30]. That is, with some small number of training iterations, the BERTAdam lacks stability optimizations, and the bias correction of Adam and AdamW is more fitting. By replacing BERTAdam with Adam and AdamW and integrating different causal networks in the EREC model, the effect of the downstream task COPA improved to varying degrees. It is worth noting that the combination of *CNT/Line* and *AdamW* achieved an accuracy of 82.6%, which is 7 points higher than BERT-base. This also confirms the effect of the model EREC proposed in this paper again.

### 5.4. Story Ending Prediction

In order to check the effectiveness of the EREC model for other tasks, we further apply the model to the Story Cloze Test (SCT) [8]. The task of SCT is to select the right ending from two candidate endings (one is incorrect and the other is correct) given a story context. There is a strong similarity between this task and COPA, where the core is the prediction of the next sentence, so we believe that applying the EREC model to the SCT task in a similar fashion will also bring some degree of enhancement. According to the processing of SCT data in [2], SCT_v1.5 [31] is used for experiments. The segmentation of SCT_v1.5 and the data volume of each part are shown in Table 5.

**Table 5.** Statistics of the datasets used in our experiments.

| Dataset | Training | Development | Test |
|---------|----------|-------------|------|
| SCT_v1.5 | 1871 | 1571 | 1571 |

We use only a 6K short news corpus of the NYT Data of the Gigaword Corpus used in [16], in which "NYT Data" refers to the textual corpus of *The New York Times*. We preprocess the data according to the process described in Section 3.2. In the second stage of incremental training, we use the test data (SCT-test) in SCT_v1.5 as the evaluation data, and when it reaches the optimal effect, the model's parameters are saved. Finally, the model is fine-tuned to solve SCT with new parameters. In the fine-tuning stage, we follow the point of [30], and the AdamW optimizer is used; then, multiple rounds of training are conducted.

As Table 6 shows, the score of the EREC model on the SCT_v1.5 test datasets reached 90.4%, which is 2.2% higher than BERT-base and slightly higher than *TransBERT*. *TransBERT* refers to $BERT_{LARGE} + MNLI$ mentioned in [2]. The results show that extra knowledge helps the model make full use of small training data and the EREC model that integrates event chains has a deeper understanding of the event relationship and causal relationship contained in the corpus, which is important for most NLP tasks as large-scale annotated data are unavailable. Thus, it can be used in the story prediction task to improve results.

**Table 6.** Experimental results of story ending prediction on the SCT_v1.5 test dataset.

| Method | Accuracy (%) |
|--------|--------------|
| BERT-base (Our Implementation) | 88.2 |
| TransBERT [2] | 90.3 |
| EREC (Ours) | **90.4** |

## 6. Conclusions

In this paper, based on BERT and ERNIE, we propose a model that integrates the event relationship and causality contained in the corpus, named enhanced language representation with event chains (EREC). A three-stage process design is used, and a variety of comparative experiments are performed in downstream tasks. Experimental results show that EREC achieves good results in causal reasoning tasks and story prediction tasks. In addition, the model uses knowledge enhancement to solve specific tasks, and the training cost is relatively low, which can effectively solve the problem of an insufficient number of samples and improve the model's utilization of prior knowledge in the field. Event induction enables the model to perform better when dealing with unknown events and is, therefore, more practical.

The main directions of future work could include the following:

(1) The model proposed in this paper will be used to conduct experiments on more domain-related and task-related corpus of other NLP tasks to further broaden its scope of application.

(2) We aim to further improve the accuracy and variety of event chains extracted from the corpus, including other forms of keywords that are not limited to verbs.

In addition, we will further optimize the model's structure and process design in future, and we hope to continuously update new events at a lower cost to promote the integration and development of language models and external knowledge.

**Author Contributions:** Conceptualization, H.W. and Y.W.; methodology, H.W.; software, H.W.; validation, H.W.; formal analysis, H.W.; investigation, H.W.; resources, H.W. and Y.W.; data curation, H.W.; writing—original draft preparation, H.W.; writing—review and editing, H.W. and Y.W. All authors have read and agreed to the published version of the manuscript.

**Funding:** This work was supported by the National Natural Science Foundation of China (under Project No. 61375053).

**Institutional Review Board Statement:** Not applicable.

**Informed Consent Statement:** Not applicable.

**Data Availability Statement:** The *CausalBank* data presented in this study are available in [20].

**Conflicts of Interest:** The authors declare no conflict of interest.

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
