# Peer review of "EREC: Enhanced Language Representations with Event Chains"

_information, doi:10.3390/info13120582_

Round 1
Reviewer 1 Report
This paper proposed a new method to improve the accuracy of a natural language representation and showed the advantage of the proposed method by several experiments. But the proposed method seems to be complex structure as shown in Figure 1, I am afraid that processing cost and time is increasing for a large number of language representaion.
Additionally, I have below small comments.
- I could not understand "qD" in line 147 , "5e-5" in line 302 and "NYT Data" in line 404.
- Equation number will be necesssary between line 153 and 154.
- In line 424, the authors said " the training cost in ralatively low", but I coul not find the reason of the opinion.
Reviewer 2 Report
This paper propose an Enhanced language Representation with Event Chains (EREC), which focuses on keywords in the text corpus and their implied relations. Event chains are integrated into EREC as external knowledge. Furthermore, various graph networks are used to generate embedding and associate keywords in the corpus. Finally, through multi-task training, the external knowledge is integrated into the model generated in the pre-training stage. The experimental process of EREC model is carried out with a three-stage design and the experimental results show that EREC has a deeper understanding of the causal relationship and event relationship contained in the text through integrating the event chains, and achieves significant improvements on two specific tasks.
The paper is interesting and within the scope of MDPI Information Processes. However some observations have to be done on this paper as follows:
1- The natural language model BERT (define BERT) need to be explained in the early section.
2- Some extensive results from the comparisons need to be added in Section 5.3. (Table 3) and Table 4.
3- In Section 5.4., missing critical discussions on the findings.
4- The following references should be included in the paper to strengthen the findings of the research works:
SPOT: Knowledge-Enhanced Language Representations for Information Extraction. CIKM 2022: 1124-1134
Multilevel Image-Enhanced Sentence Representation Net for Natural Language Inference. IEEE Trans. Syst. Man Cybern. Syst. 51(6): 3781-3795 (2021)
Multi-modal Sign Language Recognition with Enhanced Spatiotemporal Representation. IJCNN 2021: 1-8
Using Augmented Virtual Reality to Improve English Language Learning. SoMeT 2018: 759-770
- Word-length algorithm for language identification of under-resourced languages. J. King Saud Univ. Comput. Inf. Sci. 28(4): 457-469 (2016)
Cross-lingual sentiment classification using multiple source languages in multi-view semi-supervised learning. Eng. Appl. Artif. Intell. 36: 195-203 (2014)
Reviewer 3 Report
The paper focuses on an interesting topic and presents a decent solution that is supported by the needed experiments.
The related work can be extend so as to better cover the related literature.
For example, consider including the following references:
- Xiaozhou Li et al. A Sentiment-Statistical Approach for Identifying Problematic Mobile App Updates Based on User Reviews. Information, vol. 11, no. 3, 2020.
- Xiaozhou Li et al. Mobile App Evolution Analysis based on User Reviews. SOMET 2018.
Reviewer 4 Report
The authors describe the EREC model, the core of the contribution includes modules of preprocessing, corpus, extracting event chains, aligning text, fusing graph network representations, and incremental training.
The methods and experiments in the present paper are clearly explained, interesting results are obtained. Some examples and explanations may be useful to the understanding of some terms and tasks mentioned in the text, such as , the commonsense causal reasoning and story ending prediction tasks, the Choice of Plausible Alternatives (COPA), the Story Cloze Test (SCT).
The procedure to produce noise could be further explained. The example provided in line 140 could be changed to one related to the topic of the paper.
Error analysis is useful to improve the understanding of the limits of the approach.
Finally, are the the code sources available with an open licence? what about a data paper?
